# Progerinin, an Inhibitor of Progerin, Alleviates Cardiac Abnormalities in a Model Mouse of Hutchinson–Gilford Progeria Syndrome

**DOI:** 10.3390/cells12091232

**Published:** 2023-04-24

**Authors:** So-mi Kang, Seungwoon Seo, Eun Ju Song, Okhee Kweon, Ah-hyeon Jo, Soyoung Park, Tae-Gyun Woo, Bae-Hoon Kim, Goo Taeg Oh, Bum-Joon Park

**Affiliations:** 1Department of Molecular Biology, College of Natural Science, Pusan National University, Busan 46231, Republic of Korea; rosa.somi.kang@hanmail.net (S.-m.K.);; 2Heart-Immune-Brain Network Research Center, Department of Life Science, Ewha Womans University, Seoul 03760, Republic of Korea; 3Imvastech, 305, Science Building C, Ewha Womans University, Seoul 03760, Republic of Korea; 4PRG S & Tech Inc., Geumjeong-gu, Busan 46274, Republic of Korea

**Keywords:** Hutchinson–Gilford Progeria Syndrome (HGPS), progerin, cardiac dysfunction, fibrosis, *Lmna^G609G^* model mouse, Progerinin

## Abstract

Hutchinson–Gilford Progeria Syndrome (HGPS) is an ultra-rare human premature aging disorder that precipitates death because of cardiac disease. Almost all cases of HGPS are caused by aberrant splicing of the *LMNA* gene that results in the production of a mutant Lamin A protein termed progerin. In our previous study, treatment with Progerinin has been shown to reduce progerin expression and improve aging phenotypes in vitro and in vivo HGPS models. In this record, cardiac parameters (stroke volume (SV), ejection fraction (EF), fractional shortening (FS), etc.) were acquired in *Lmna^WT/WT^* and *Lmna^G609G/WT^* mice fed with either a vehicle diet or a Progerinin diet by echocardiography (from 38 weeks to 50 weeks at various ages), and then the cardiac function was analyzed. We also acquired the tissue samples and blood serum of *Lmna^WT/WT^* and *Lmna^G609G/WT^* mice for pathological analysis at the end of echocardiography. From these data, we suggest that the administration of Progerinin in the HGPS model mouse can restore cardiac function and correct arterial abnormalities. These observations provide encouraging evidence for the efficacy of Progerinin for cardiac dysfunction in HGPS.

## 1. Introduction

Hutchinson–Gilford Progeria Syndrome (HGPS) is a rare genetic disorder caused by an autosomal dominant mutation in the *LMNA* gene resulting in the accumulation of a truncated Lamin A version called progerin. Accumulation of progerin causes nuclear dysfunction and morphological abnormalities [1,2,3]. Patients with HGPS develop numerous clinical features including short stature, alopecia, scleroderma-like skin, low bone mineral density, lipodystrophy, skeletal dysplasia, and strokes. Premature death usually occurs in the second decade of age because of cardiovascular complications [4,5,6,7]. Transgenic *Lmna^G609G^* knock-in mice display phenotypes characterized by somatic growth, kyphosis, mandible hypoplasia, malocclusion, skin and adipose tissue atrophy, cardiovascular alterations, and shortened life span observed in patients with HGPS [8,9].

Previous studies have demonstrated that progerin strongly binds to wild-type Lamin A leading to disruption of nuclear architecture and acceleration of premature aging [10]. Pharmacological inhibition of Lamin A-progerin interaction by Progerinin (SLC-D011) improves cellular phenotypes in HGPS fibroblasts. Furthermore, *Lmna^G609G^* progeroid mouse model has shown the ability of Lamin A-progerin binding inhibitor treatment to improve phenotypes [11]. However, the effect of Progerinin on cardiovascular function and phenotypes has not been sufficiently proven. Given that premature cardiovascular disease is the cause of death in most of the patients with HGPS, we sought to test the ability of Progerinin to reduce the cardiovascular alterations seen in transgenic *Lmna^G609G^* mice.

## 2. Materials and Methods

### 2.1. Mice

All procedures with mice were performed in a facility certified by the Association for Assessment and Accreditation of Laboratory Animal Care in compliance with animal policies approved by Ewha Womans University. The mouse work was performed under the study protocol (approval code: 19-042), as approved by the Institutional Animal Care and Use Committee. Male heterozygous *Lmna^G609G^* mice, provided by Carlos López-Otín (Universidad de Oviedo, Asturias, Oviedo, Spain), and male wild-type C57BL/6J mice were housed at the specific pathogen-free facility.

### 2.2. Progerinin (SLC-D011) Synthesis and Charaterization Data

The synthetic scheme of Progerinin has already been described in our previous study [11]. In brief, Progerinin was synthesized by removing the hydrogen from the hydroxy of the decursinol which was isolated from the roots of Angelica gigas Nakai and by attaching side chains. After synthesis, the unrefined Progerinin was completed as the final compound by recrystallization. The compound was synthesized in a batch (Batch No. A05064-013S2; Batch size: 23.5 g, which was the same batch used in our previous study). The purity of Progerinin was measured by HPLC (≥97.5%). The molecular structure of Progerinin was investigated using H-NMR Spectroscopy, C-NMR spectroscopy, mass spectrometry, and X-ray diffractometry [11].

### 2.3. Chemical Pharmacokinetic (PK) Analysis

PK analysis of Progerinin has already been described in the previous study [11]. Briefly, 10 mg/kg of Progerinin in 10% NMP and 90% PEG400 solution were orally delivered. At pre-set time points, the blood concentration of Progerinin was determined by LC-MS/MS analysis.

### 2.4. Preparation of Feed Pellets

For oral administration of Progerinin by diet pellets, Progerinin was suspended in a monoolein-based solution at a concentration of 10 mg/mL. The base solution was made of monoolein (Peceol^TM^ (code: 3088BFC, Lot/Batch: 174894), Gattefossé, La Défense Cedex, France) and tricaprylin (Labrafac^TM^ lipophile WL 1349 (code: 3139BFC, Lot/Batch: 171801), GATTEFOSSE, Lyon, France). The suspensions were heated (50 °C) and sonicated continuously to make a clear solution. A powdered diet was mixed with the clear monoolein-based solution consisting of Progerinin and was made into dough balls. The vehicle-group diet was made of the clear monoolein-based solution without Progerinin. The dough balls were stored at 4 °C until use.

### 2.5. Progerinin Treatment

Progerinin was administrated with the same dose as in the previous study [11]. Mice were orally administrated with Progerinin-dough balls (2 g/day) daily at a concentration of 50 mg/kg. Control mice were treated with vehicle-dough balls in the same way. Mice were fed with a vehicle diet or therapeutic Progerinin diet starting from 38 weeks of age and sacrificed at 50 weeks of age.

### 2.6. Echocardiography and Cardiac Function Analysis

Mouse-specific ultrasound Vevo2100 system used to analyze and evaluate cardiac function and myocardial changes in detail using software such as LV analysis, PW Tissue Doppler Mode, and VEVO strain. Echocardiography was performed after general anesthesia using the inhaled anesthetic agent isoflurane (2–2.5% and 0.5 L/min oxygen) and removal of the hair coat of the whole chest. While positioning the mouse at an angle to the right, axis images of the left ventricle from the central thoracic and M-mode ultrasound images at the papillary muscle level were obtained. Cardiac function was quantified by analyzing cardiac function parameters. Before feeding and at various time points after feeding, in vivo cardiac parameters were obtained by the images from echocardiography and calculated using the following formula: HR (bmp) = total heart beats*60/time; SV (µL) = EDV − ESV; CO (mL/min) = SV*HR; LV EF (%) = (LVEDV − LVESV)/LVEDD*100%; LV FS (%) = (LVEDD − LVESD)/LVEDD*100%; DWS (%) = (LVPWs − LVPWd)/LVPWs*100%.

### 2.7. Western Blot for Tissue Samples

Progerin amounts in heart tissue analysis of wild-type mice, vehicle-treated *Lmna^G609G/WT^* mice, and Progerinin-treated *Lmna^G609G/WT^* mice were evaluated by means of Western blot analysis. Proteins were extracted with RIPA lysis buffer supplemented with protease inhibitors. Tissues were minimized by a tissue grinder and centrifuged at 15,000 rpm for 15 min at 4 °C. Proteins were loaded onto SDS-PAGE gels and transferred to the PVDF membranes. These membranes were blocked with 3% skimmed milk in TBS-T buffer for 1 h and incubated with primary antibodies, including mouse monoclonal anti-progerin (1:100; sc-81611; Santa Cruz Biotechnology, Dallas, TX, USA) and mouse monoclonal anti-beta actin (1:10,000; 66009-1-Ig; Proteintech, Rosemont, IL, USA) at 4 °C overnight. Blots were then incubated with goat anti-mouse IgG (1:10,000; Thermo Fisher Scientific, Waltham, MA, USA) secondary antibody and washed with TBS-T washing buffer. Bands were developed using the Intron ECL detection system and quantified using Image J (version: 2.9.0/1.53t, National Institute of Health (NIH), Bethesda, MD, USA).

### 2.8. Histology Analysis

Tissue specimens, including the heart and ascending aorta, were fixed in 4% PFA and embedded in paraffin. Paraffin blocks were sectioned and transferred onto adhesive-coated slides. After deparaffinization and rehydration, tissue sections were stained with H&E (hematoxylin and eosin), Verhoeff-Van Gieson, or Masson Trichrome. Immunohistochemistry was carried out using primary antibodies including anti-αSMA (1:3000; 14395-1-AP; Proteintech, Rosemont, IL, USA) and anti-Lamin A/C (1:100; 10298-1-AP; Proteintech, Rosemont, IL, USA). The images of the aorta stained with Masson Trichrome were separated into three different colors (blue, red, and green) by color deconvolution of ImageJ/FIJI to analyze the composition of collagen (blue) and muscle tissue (red). The number of elastic fiber breaks was measured in a single aortic ring from different individuals. The percentage of abnormal nuclei in heart tissue was calculated based on the manual counts carried out on randomly selected individuals in the same area of each group. Abnormalities of nuclei were determined based on extruded or engulfed nuclear shape, and irregular contour. For identifying clear images of nuclei, color deconvolution by ImageJ/FIJI was applied to the IHC images, which were stained with the anti-Lamin A/C antibody.

### 2.9. Mouse Cytokine Array

Whole blood samples of mice were collected after sacrifice at 50 weeks of age. The whole blood was left to clot at room temperature. Serum was collected by removing the clot following centrifugation at 4 °C. Cytokine array was performed using Proteome Profiler Mouse Cytokine Array Kit (ARY006; R&D systems, Minneapolis, MN, USA) following the manufacturer’s instructions. The array images were quantified using HLImage++ software 6.2 (Ideal Eyes Systems, Bountiful, UT, USA).

### 2.10. Statistical Analysis

Statistical analyses were performed using GraphPad Prism (Version 9.0.0, San Diego, CA, USA). Data were represented as means ± standard error mean (SEM) or standard deviation (SD). The statistical significance was analyzed using a one-way ANOVA followed by Tukey’s test or a two-way ANOVA Fisher’s LSD test.

## 3. Results

### 3.1. Progerinin Improves Cardiac Dysfunction in Lmna^G609G/WT^ Mice

To measure cardiac physiology and architecture in HGPS mice, we examined cardiac function in 38- to 50-week-old *Lmna^G609G/WT^* mice using echocardiography. Progerinin mixed in *Lmna^G609G^* mouse dough or vehicle alone was orally administered at 50 mg/kg/day, which has been shown previously to be an effective dose to reduce progerin level in the HGPS mouse model (Appendix A). Progerinin intake delayed the reduction in body weight in *Lmna^G609G/WT^* mice compared to the vehicle group (Appendix A). In vivo, cardiac parameters by echocardiography revealed that HGPS mice displayed systolic and diastolic dysfunction compared to wild-type mice (Appendix A). Echocardiography revealed no significant between-diet differences in heart rate (HR) (Appendix A). Comparing the stroke volume of *Lmna^WT/WT^* mice with *Lmna^G609G/WT^* mice before and after feeding, stroke volume was remarkably reduced in *Lmna^G609G/WT^* mice fed with the control diet (Figure 1A). However, significant improvements were confirmed from 8 weeks (46-week-old) in the Progerinin-fed group. There was no significant difference in cardiac output between the vehicle group and the Progerinin-fed group (Appendix A). As the most common measure of cardiac function, ejection fraction (EF) is the ratio of blood flow that has been released from the left ventricle during the heart cycle (EF = Cardiac output/Left ventricle volume at diastolic end*100%). The *Lmna^G609G/WT^* mice displayed overt left ventricle systolic dysfunction, with a significantly lower ejection fraction and fractional shortening (FS) than *Lmna^WT/WT^* mice. However, the Progerinin-fed *Lmna^G609G/WT^* group showed improvement in systolic function after 8 weeks (Figure 1B,C).

We next analyzed the function of left ventricle relaxation. The *Lmna^G609G/WT^* mice showed a reduced level of left ventricular posterior wall thickness at end-systole (LVPWs) and end-diastole (LVPWd). Comparing the LVPWs and LVPWd between the vehicle group and the Progerinin-fed group, *Lmna^G609G/WT^* mice showed no outstanding difference. Diastolic wall strain (DWS) has recently been used as a left ventricle function marker, indicating the relaxation of the left ventricle (DWS = [LVPWs − LVPWd]/LVPWs × 100 (%)). Although the difference in DWS between *Lmna^WT/WT^* mice and *Lmna^G609G/WT^* mice in the vehicle group was not significant, it showed a declining trend. Comparing the DWS between the vehicle group and the Progerinin-fed group, *Lmna^G609G/WT^* mice showed remarkable differences in long-term feeding, even though the DWS variation between mice is large (Figure 1D). These results demonstrate the effective improvement of cardiac function of HGPS mice by the long-term treatment of Progerinin.

### 3.2. Progerinin Reduces Histological Aortic Defects in Lmna^G609G/WT^ Mice

Next, we performed microscopy studies in the aorta and heart to investigate the histological alteration. Arterial sections obtained after necropsy were analyzed by Masson Trichrome (MT) staining which is used in histology to differentiate collagen and muscle fibers on tissue sections. To simplify the complex staining and overlapping stains, images of aortic lesions were separated by using ‘color deconvolution’ of ImageJ/FIJI. MT staining in the aorta revealed significantly higher collagen decomposition (blue) in HGPS mice, accompanied by loss of muscle tissue (red). However, treatment with Progerinin reduced aortic fibrosis by restoring the composition of collagen and muscle tissue as in wild-type mice (Figure 2A and Appendix A). To visualize elastic fibers in the aorta, the sections from the three groups were stained with Verhoeff-Van Gieson’s stain. Verhoeff-Van Gieson staining in the aorta of *Lmna^G609G/WT^* mice and showed an increased number of elastin fiber breaks. However, Progerinin intake decreased the number of elastin fiber breaks (Figure 2B,C and Appendix A). Loss of smooth muscle cells is a hallmark of HGPS vessels. We also observed a lack of vascular smooth muscle cells (VSMCs) by α-smooth muscle actin staining in the aorta in *Lmna^G609G/WT^* mice. However, compared with the vehicle *Lmna^G609G/WT^* mouse group, the composition of smooth muscle cells was remarkably restored in the Progerinin-fed group (Appendix A).

### 3.3. Progerinin Reduces Nuclear Deformation and Progerin Expression in Heart Tissue of Lmna^G609G/WT^ Mice

Nuclear deformation is a hallmark of HGPS [5,12,13]. Consistent with these findings, immunochemistry studies demonstrated severely deformed nuclei in heart tissues in *Lmna^G609G/WT^* mice. However, deformed nuclear shapes in heart tissues were restored following the oral administration of Progerinin (Figure 3A and Appendix A). To identify abnormal nuclei clearly, images of heart lesions were digitally separated by ImageJ/FIJI. Quantification of abnormal nuclei in heart tissue also shows that Progerinin prevents nuclear deformation in *Lmna^G609G/WT^* mice (Appendix A).

Next, in vivo, Progerinin activity was judged by Western blots of progerin. The graph reveals a decrease in progerin protein in the heart tissues of *Lmna^G609G/WT^* mice by more than 50% after treatment with Progerinin (Figure 3B). This result supports that cardiac dysfunction and histological aberration were restored by the reduction in progerin levels after treatment with Progerinin.

### 3.4. Progerinin Reverses the Levels of Cytokines in Serum of Lmna^G609G/WT^ Mice

After 3-month treatments, we analyzed 40 different proteins in blood serum by mouse cytokine array kit to explore Progerinin’s potential impact on the levels of cytokines and chemokines which are related to cardiovascular function. We randomly selected three different mice from each group of *Lmna^WT/WT^*, *Lmna^G609G/WT^* vehicle, and *Lmna^G609G/WT^* treated mice. Two of 40 cytokines were significantly altered in *Lmna^G609G/WT^* mice compared to the wild-type model, including IL-1ra (IL-1 receptor antagonist) and TIMP-1 (Tissue inhibitor of matrix metalloproteinase-1) (Appendix A). A low level of IL-1ra in *Lmna^G609G/WT^* mice was significantly upregulated after the Progerinin intake (Figure 4A). Conversely, TIMP-1 was a high level of cytokines in *Lmna^G609G/WT^* mice and was significantly down-regulated by Progerinin intake (Figure 4B). IL-1ra counter-regulates IL-1β, a key cytokine in the development of cardiovascular diseases [14,15]. Circulating IL-1ra expression is decreased in aged C57BL/6 mice [16]. Expression of IL-1ra was strongly reduced in blood serum from *Lmna^G609G/WT^* mice compared to wild-type mice of the same age (Figure 4A). However, IL-1α and IL-1β levels were not detectable. TIMP-1 is consistently upregulated in myocardial fibrosis and is used as a marker of fibrosis [17,18]. Expression of TIMP-1 was significantly increased in *Lmna^G609G/WT^* mice. After a 3-month Progerinin intake, elevated serum TIMP-1 level was markedly reduced (Figure 4B). Collectively, these data support that the Progerinin-mediated pharmacological inhibition of progerin ameliorates myocardial fibrosis and dysfunction in the HGPS mouse model.

## 4. Discussion

We have shown the efficiency of Progerinin on classical HGPS patient-derived fibroblasts and *Lmna^G609G^* mice: Progerinin leads to progerin degradation by interrupting Lamin A-progerin interaction and rejuvenates the aging phenotypes of HGPS fibroblasts, which are the increased-senescence and the decreased-proliferation. Oral administration of Progerinin to HGPS model mice improved progeroid phenotypes including life span expansion and reduced progerin expression levels in several organs including the lungs, liver, and kidneys [11]. We, therefore, hypothesized that Progerinin could also have a beneficial impact on cardiac function and phenotypes of *Lmna^G609G/WT^* mice.

Premature cardiovascular disease is a major cause of death in HGPS patients [19,20]. It is known that progerin causes extensive atherosclerosis and cardiac electrophysiological alterations that invariably result in premature aging and death [20,21,22,23,24,25]. The *Lmna^G609G^* knock-in mouse model has histological signs of cardiovascular alterations [9,26]. The progeroid signs were more evident and severe in *Lmna^G609G/G609G^* than in *Lmna^G609G/WT^* mice [9]. However, in this study, all experiments were performed by using *Lmna^G609G/WT^* mice because of the difficulty in producing enough homozygous individuals at the same time. Here, we provide a relevant analysis of echocardiographic abnormalities and histological changes in *Lmna^G609G/WT^* mice and show the efficacy of Progerinin that the changes are restored by its treatment. Investigation of HGPS cardiac dysfunction with the *Lmna^G609G^* model reveals a reduction in stroke volume, cardiac output, and diastolic function of the left ventricle. The echocardiographic evaluation showed overall reduced systolic function in *Lmna^G609G/WT^* mice compared to *Lmna^WT/WT^* mice. The small body size of *Lmna^G609G/WT^* mice can reduce systolic functions because the change in body weight is related to systolic blood pressure [27,28]. Although Progerinin shows no significant increase in weight gain, it helps maintain body weight longer and improves systolic function in *Lmna^G609G/WT^* mice. In addition, in the case of humans, there are several ways to evaluate the relaxation function of the left ventricle system [29,30]. However, in the case of mice, it is difficult to measure the relaxation function of the left ventricular system because the heart is smaller than a human heart, and the heart rate is very high [31,32]. However, changes in DWS calculated by LVPWs and LVPWd measurements are meaningful in mice in evaluating the diastolic function of the left ventricle. By the long-term treatment of Progerinin, the key cardiac function indicators such as stroke volume, ejection fraction, and fractional shortening are improved without showing side effects.

Myocardial fibrosis is essential to cardiac remodeling, leading to heart failure and premature death. Excessive deposition of the extracellular matrix such as collagen induces myocardial fibrosis [33,34]. The elastic fibers, made of elastin and fibrillin-containing microfibrils, have an important role in providing elasticity and resilience to the tissues. Histologically, aging phenotypes in elastic fibers are characterized by thinning and fragmentation of elastic structures [35]. In our previous study, we showed that myocardial fibrosis was increased in the heart of *Lmna^G609G/WT^* mice compared to wild-type mice. However, fibrosis in heart tissue was reduced by treatment with Progerinin in *Lmna^G609G/WT^* mice [11]. Here, we show a significant increase in collagen composition and elastic fiber breaks in the aorta of *Lmna^G609G/WT^* mice. Interestingly, Progerinin improves aortic fibrosis and loss of elastic fiber integrity to similar levels as wild-type mice. The reduction in smooth muscle cells in vessels is associated with aging and HGPS [23,24,36,37]. HGPS patients who had died of myocardial infarction had severe depletion of vascular smooth muscle cells (VSMCs) in the aortic media. The patients’ VSMCs were replaced by collagen fibrils increasing arterial fibrosis [26]. The *Lmna^G609G^* model, used in our study, also shows prominent VSMC depletion. We observed a significant loss of α-smooth muscle actin in the aorta of *Lmna^G609G/WT^* mice at 50 weeks of age and these phenotypes were recovered by Progerinin treatment (Appendix A). These histological observations reinforce the findings that *Lmna^G609G/WT^* mice have progressive cardiac defects compared to wild-type mice.

Several strategies have been developed to treat HGPS, targeting the abnormal splicing of progerin, its farnesylation, clearance, or downstream signals [7,38,39,40,41]. Progerinin targets the progerin protein directly, and it inhibits the pathological interaction between Lamin A and progerin. Finally, the interruption of the binding between the two proteins makes the progerin level clear in human fibroblasts and several organs of *Lmna^G609G^* mice [11]. However, in heart tissue, the progerin level and its clearance by Progerinin treatment have not been examined yet. Therefore, it was evaluated whether Progerinin could reduce the levels of progerin and confirmed that its oral administration to *Lmna^G609G/WT^* mice efficiently reduces the level of progerin in the heart tissue (Figure 3B,C). This suggests that a significant improvement in cardiac dysfunction and histological alterations might be mediated by the clearance of progerin in heart tissues.

Interestingly, Progerinin could also reduce the level of TIMP-1 and induce the level of IL-1ra in the blood serum of *Lmna^G609G/WT^* mice (Figure 4). TIMP-1 is consistently upregulated in myocardial fibrosis and is used as a marker of fibrosis [17,42,43]. Myocardial fibrosis, overaccumulation of the extracellular matrix fibrillar collagens, is one of the main features of various cardiomyopathies and accommodates cardiac systolic and diastolic performance [33,34,43]. Therefore, we suggest that the alteration of TIMP-1 levels may be related to cardiac dysfunction and excess collagen content in the aortic wall in HGPS mice. On the other hand, IL-1ra, a natural regulator of interleukin 1 (IL-1) cytokines, is mechanistically linked to cardiovascular risk. IL-1ra blocks signals of IL-1β, a potent proinflammatory cytokine that is implicated in the development of chronic inflammatory disorders such as type 2 diabetes, rheumatoid arthritis, and cardiovascular diseases [14,44,45]. However, the relationship between IL-1ra and cardiovascular defects remains incompletely understood [46]. In our present study, we observed that the reduction in IL-1ra levels in *Lmna^G609G/WT^* mice is recovered by Progerinin intake, although we could not detect other IL-1 cytokines (Appendix A). It might be suggested from our results that low levels of IL-1ra in HGPS model mice could be related to cardiac dysfunction. Progerinin may have a favorable impact on the IL-1 system and cardiovascular events.

In this report, we have provided evidence that Progerinin intake has therapeutic potential for HGPS-related cardiovascular dysfunction and histological defects by targeting progerin suggesting that the cardiovascular disease of HGPS patients can be improved. Several studies show that other therapeutic approaches also improve cardiac phenotypes in progeria model mice [47,48,49,50]. Especially, lonafarnib, the only drug currently in the market approved for HGPS treatment, improves arterial structure, and left ventricular diastolic function [51]. Like lonafarnib, Progerinin also shows favorable effects on cardiac function and phenotypes. However, in our previous study, we showed that treatment with Progerinin is more non-toxic and effective than lonafarnib [11]. Along with the previous study, the present findings may imply that Progerinin treatments could be better at reducing the risk of premature death in HGPS.

## 5. Conclusions

The evaluation of in vivo cardiac parameters by echocardiography reveals that Progerinin diet-fed *Lmna^G609G/WT^* mice have significant improvement in the key cardiac function including stroke volume, ejection fraction, and fractional shortening compared with vehicle diet-fed *Lmna^G609G/WT^* mice. Moreover, cardiovascular phenotypes of *Lmna^G609G/WT^* mice are ameliorated by Progerinin intake. These results demonstrate that Progerinin has a possibility of therapeutic strategy on the cardiovascular system in HGPS patients.

## Figures and Tables

**Figure 1 cells-12-01232-f001:**
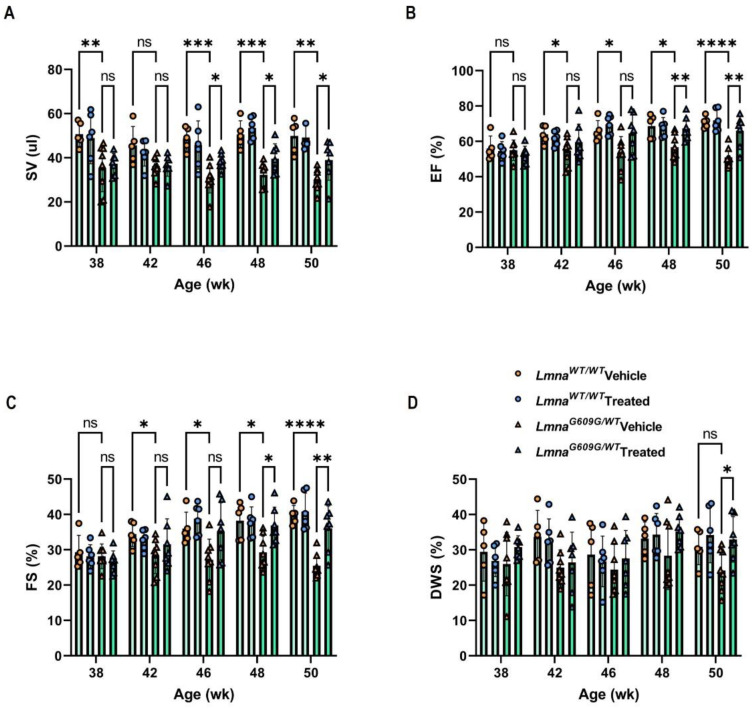
Analysis of cardiac function in HGPS model mice by therapeutic drug, Progerinin. Cardiac parameters of stroke volume (SV; (**A**)); ejection fraction (EF; (**B**)); fractional shortening (FS; (**C**)); and diastolic wall strain (DWS; (**D**)) measured at 38, 42, 46, 48, and 50 weeks of age in the vehicle and Progerinin-treated wild-type mice and *Lmna^G609G/+^* mice (*Lmna^WT/WT^* Vehicle: *n* = 5; *Lmna^WT/WT^* Treated: *n* = 6; *Lmna^G609G/WT^* Vehicle: *n* = 8; *Lmna^G609G/WT^* Treated: *n* = 7). Statistical analysis was performed using two-way ANOVA followed by Fisher’s LSD test. * *p* < 0.05, ** *p* < 0.01, *** *p* < 0.001, **** *p* < 0.0001, and ns: not significant.

**Figure 2 cells-12-01232-f002:**
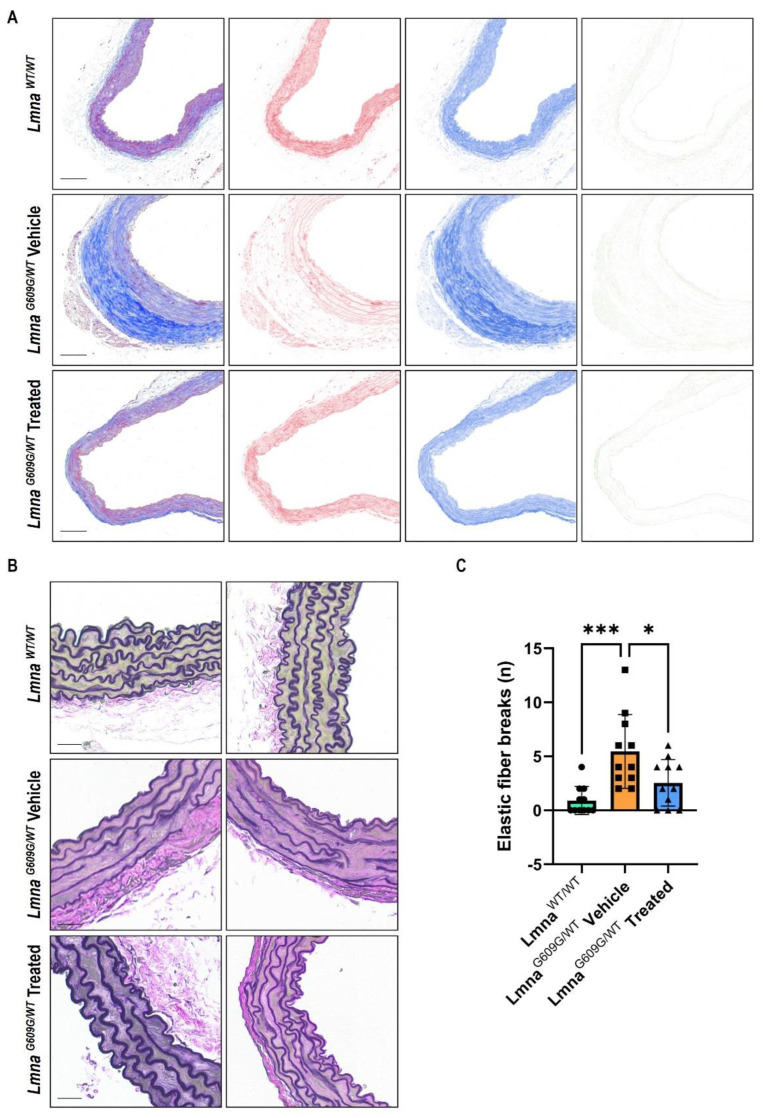
Improvement of aortic fibrosis and elastic fiber breaks by Progerinin in *Lmna^G609G/WT^* mice. (**A**) Masson’s trichrome staining of the aorta of wild-type mice and *Lmna^G609G/WT^* mice. Images of aortic lesions were digitally separated into three stains (red, blue, and green) by using ‘color deconvolution’ of ImageJ/FIJI. The blue indicates collagen decomposition, and the red indicates muscle tissue. Progerinin restored the composition of collagen and muscle tissue in *Lmna^G609G/WT^* mice. The scale bar represents 100 µm. Images of Verhoeff-Van Gieson staining (**B**) and the number of elastic fiber breaks (**C**) in the aorta of wild-type mice and *Lmna^G609G/WT^* mice (*Lmna^WT/WT^*: *n* = 11; *Lmna^G609G/WT^* Vehicle: *n* = 11; *Lmna^G609G/WT^* Treated: *n* = 11). The scale bar represents 25 µm. Statistical analysis was performed using one-way ANOVA followed by Tukey’s test, * *p* < 0.05, *** *p* < 0.001.

**Figure 3 cells-12-01232-f003:**
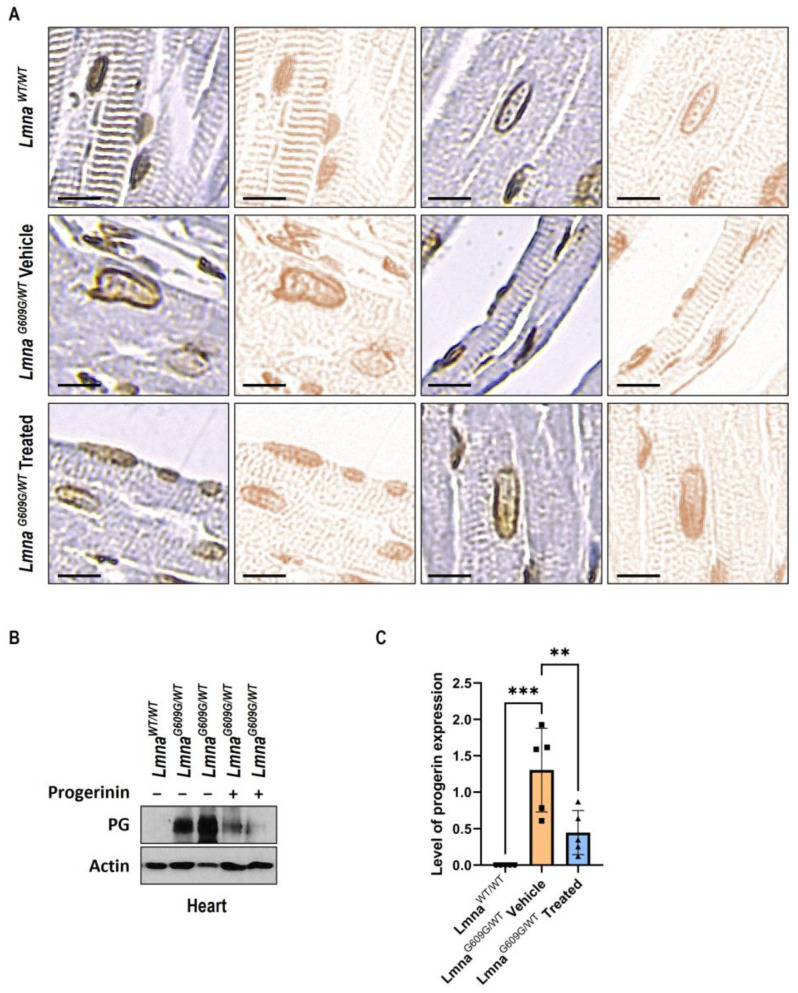
Nuclear deformation in heart tissue is prevented by Progerinin intake. (**A**) Immunohistochemistry (IHC) of heart tissue samples stained with an anti-lamin A/C antibody (*Lmna^WT/WT^*: *n* = 8; *Lmna^G609G/WT^* Vehicle: *n* = 8; *Lmna^G609G/WT^* Treated: *n* = 8). Images of heart lesions were separated by color deconvolution. The scale bar represents 10 µm. (**B**) Western blot analysis of progerin in the heart tissues of wild-type mice and *Lmna^G609G/WT^* mice. (**C**) Quantitative analysis of progerin expression in the heart tissues. Progerinin intake reduces the levels of progerin expression in *Lmna^G609G/WT^* mice (*Lmna^WT/WT^*: *n* = 5; *Lmna^G609G/WT^* Vehicle: *n* = 5; *Lmna^G609G/WT^* Treated: *n* = 5). Statistical analysis was performed using one-way ANOVA followed by Tukey’s test, ** *p* < 0.01, *** *p* < 0.001.

**Figure 4 cells-12-01232-f004:**
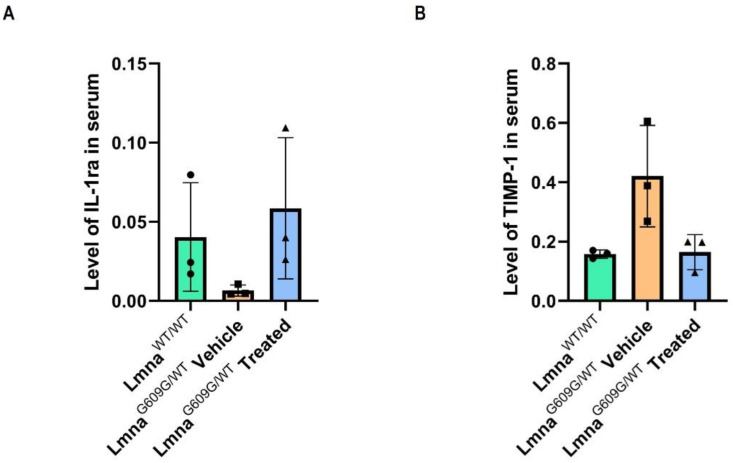
Analysis of cytokines in blood serum by Progerinin treatment. The expression level of IL-1ra protein (**A**) and TIMP-1 (**B**) from blood serum of wild-type mice and *Lmna^G609G/+^* mice. Samples of blood serum were collected from 3 different mice of each group (*Lmna^WT/WT^*; *Lmna^G609G/WT^* Vehicle; and *Lmna^G609G/WT^* Treated). Statistical analysis was performed by ordinary one-way ANOVA (ANOVA summary *p* < 0.05).

## Data Availability

All data that support the findings of this study are available from the corresponding author upon reasonable request.

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
