# Peer review of "Progerinin, an Inhibitor of Progerin, Alleviates Cardiac Abnormalities in a Model Mouse of Hutchinson–Gilford Progeria Syndrome"

_cells, 2023, doi:10.3390/cells12091232_

Round 1

Reviewer 1 Report

Children with HGPS develop disease that affects multiple tissue systems but die as a result of cardiovascular disease (i.e., heart attack and stroke) in their mid-teens. Classical HGPS is caused by a mutation in LMNA resulting in the synthesis of an abnormal nuclear lamin protein named progerin. Genetic approaches to correct the LMNA HGPS mutation (e.g., base-editing) have proven to be highly effective in mouse models. However, until those approaches are translated into treatments, new medical approaches are needed. In earlier studies, the authors showed that Progerinin—an orally active compound that reduces the association of progerin with lamin A—reduces cellular phenotypes associated with progerin toxicity (e.g., abnormal nuclear shape), and extends survival in LmnaG609G/+ mice (a model of HGPS). In this paper, the authors investigate the effects of Progerinin on HGPS disease in the aorta and heart of LmnaG609G/+ mice. They report that Progerinin improves echocardiographic measures of cardiac function (SV, EF, and FS) but not cardiac output (HR was not affected in HGPS mice). The authors also report that Progerinin reduces nuclear deformation in cardiomyocytes and improves tissue morphology in the aorta (as judged by elastic fiber breaks, tissue fibrosis, and SMC numbers). The authors conclude that these results provide evidence for the efficacy of Progerinin to treat cardiac abnormalities in HGPS. 

Comments.

1. The authors make statements that are unsupported. For example, the authors state that Progerinin delayed the reduction of body weight in HGPS mice (line 111). However, in Suppl. Fig. 2, the change in body weight in the vehicle and treated groups do not look different. Similarly, the authors state that Progerinin reduces histological cardiac defects (line 144) and improves myocardial fibrosis (line 235). However, no evidence is shown to support either conclusion. Also, Progerinin was reported to reduce elastic fiber breaks in the aorta (Fig. 2C). However, statistical analysis (ANOVA) was not performed for the Progerinin and Vehicle groups; thus, it is unclear if the number of breaks is different. There are other examples.           

2. How was the echocardiographic data analyzed (Fig. 1)? It appears the groups were analyzed by t-test instead of one-way ANOVA (for each age group). Based on the variance, it is likely that some of the parameters may not be improved by Progerinin treatment (e.g., SV).

3. Previous studies have clearly shown that the severity of aortic disease in LmnaG609G/+ mice is highly dependent on location (e.g., inner vs outer curvature of the ascending aorta; ascending vs descending aorta) (Kim et al. Sci Trans Med 2018). In Fig. 2 and Suppl Fig. 4-6, were the same locations analyzed for the different groups? The site should be identified in the Methods. Importantly, is there any quantitative data to support the conclusion that Progerinin improves aortic disease? The authors did show decreased numbers of elastic fiber breaks (Fig. 2C), but how where the breaks measured? Breaks/area (how where the regions selected?) or breaks/aortic ring? In Fig. 2A and Suppl Fig. 4, what are the blank boxes?

4. Nuclear deformation was analyzed in heart tissue by IHC (Fig. 3A). Is there any quantitative data? The images are out of focus (new ones need to be collected). What is the middle panel for the western blot in Fig. 3B?

Author Response

Thank you so much for taking your time to review our paper.

  1. The authors make statements that are unsupported. For example, the authors state that Progerinin delayed the reduction of body weight in HGPS mice (line 111). However, in Suppl. Fig. 2, the change in body weight in the vehicle and treated groups do not look different. Similarly, the authors state that Progerinin reduces histological cardiac defects (line 144) and improves myocardial fibrosis (line 235). However, no evidence is shown to support either conclusion. Also, Progerinin was reported to reduce elastic fiber breaks in the aorta (Fig. 2C). However, statistical analysis (ANOVA) was not performed for the Progerinin and Vehicle groups; thus, it is unclear if the number of breaks is different. There are other examples.

→ We changed body weight data over the 12-week treatment period. The graph shows that Progerinin-treated mouse group showed noticeable difference (p<0.05, the mean values) compared with vehicle-treated mouse group.

→ There seems to have been choosing the improper terms ‘cardiac defects’ and ‘myocardial fibrosis’. We already analysed efficacy of Progerinin in heart of LmnaG609G/WT mice in our previous study thus we stated comprehensively with our previous study. In this study, we are focusing on aortic histology. So, we exchanged the word into ‘aortic defects’ and ‘aortic fibrosis’ to be clear.

→ We performed statistical analysis by using one-way ANOVA test and uploaded in Figure 2C.

  1. How was the echocardiographic data analyzed (Fig. 1)? It appears the groups were analyzed by t-test instead of one-way ANOVA (for each age group). Based on the variance, it is likely that some of the parameters may not be improved by Progerinin treatment (e.g., SV).

→We analyzed the echocardiographic data by t-test. Because, in this study, we mainly focus on analyzing cardiac function between wild type mice and progeria mice or between vehicle-fed LmnaG609G/WT mice and Progerinin-fed LmnaG609G/WT mice. Thus, we analyzed the data using t-test to compare between two groups.

→Even though the variation between mice is slight large, noticeable improvements were confirmed from 8 weeks (46 weeks old) in Progerinin-fed group as a result of comparing the stroke volumes between control diet-fed and Progerinin diet-fed group.

  1. Previous studies have clearly shown that the severity of aortic disease in LmnaG609G/+ mice is highly dependent on location (e.g., inner vs outer curvature of the ascending aorta; ascending vs descending aorta) (Kim et al. Sci Trans Med 2018). In Fig. 2 and Suppl Fig. 4-6, were the same locations analyzed for the different groups? The site should be identified in the Methods. Importantly, is there any quantitative data to support the conclusion that Progerinin improves aortic disease? The authors did show decreased numbers of elastic fiber breaks (Fig. 2C), but how where the breaks measured? Breaks/area (how where the regions selected?) or breaks/aortic ring? In Fig. 2A and Suppl Fig. 4, what are the blank boxes?

→ We are sorry for the lack of information. We collected and analyzed the ascending aorta in this study. We analyzed the same locations of aorta (the ascending aorta) from the different group. We identified the site of aorta in the methods section.

→ In this study, we could only quantify the number of elastin fiber breaks in aorta by counting the breaks. Other results were visualized by microscopic images.

→ We measured the breaks of single aortic ring from different individuals. We updated in the method section.

→ To simplify the complex staining of arterial sections, we separated the MT stained images by using ‘color deconvolution’ of ImageJ/FIJI. The images were digitally separated into three stains (red, blue, and green). The blank boxes, which you mentioned, are the separated green stains of MT staining. In fibrosis analysis, however, the red stains and the blue stains are meaningful results.

  1. Nuclear deformation was analyzed in heart tissue by IHC (Fig. 3A). Is there any quantitative data? The images are out of focus (new ones need to be collected). What is the middle panel for the western blot in Fig. 3B?

→We changed the improved images and also added a graph of percentage of abnormal nuclei in supplementary data (Supple Figure 7B).

→We are sorry for confusing you. We noticed that we made a mistake in Figure 3B. We have added two different cropped ‘PG’ bands. We have done WB repeatedly. Each of two bands is from separated experiment. We corrected Figure 3B.

We greatly appreciate the time you spent.

Reviewer 2 Report

The work about SLC-D011 effect in cardiac abnormalities is really good. However, the authors should be address some comments:

Comment 1. In the results described based on Figure 2 are really complicated to understand. Please, could you improve the explaination.

Comment 2. The quality of the Figure 3 is low. The authors should be improved.

Comment 3. The authors should be included the completed membrane of the WB in the Figure 3 and include the molecular weight

Author Response

Thank you so much for taking your time to review our paper.

Comment 1. In the results described based on Figure 2 are really complicated to understand. Please, could you improve the explanation.

→ I am sorry that our explanation is complicated to understand. Reflecting your opinion, we added some improved explanations for easy understanding.

Comment 2. The quality of the Figure 3 is low. The authors should be improved.

→We changed the improved images and also added a graph of percentage of abnormal nuclei in supplementary data (Supple Figure 7B).

Comment 3. The authors should be included the completed membrane of the WB in the Figure 3 and include the molecular weight.

→We have attached the uncropped membrane in supplementary data with the molecular weight.

We greatly appreciate the time you spent to us.

Reviewer 3 Report

So-mi Kang et al. report in this manuscript the improved cardiovascular abnormalities in progeria LmnaG609G/wt mouse model upon Progerinin (SLC-D011) treatment covering cardiac functional assessment, histopathological analysis, which most likely resulted from reduced level of toxic progerin protein and decreased inflammation. This study is of great interest in investigating the cardiac benefits of Progerinin as a potential therapy for HGPS patients, additional experiments need to be conducted and some results need to be reanalyzed to soundly support their conclusion. Outlined below are specific comments to help improve this manuscript.

1. The echocardiographic evaluation of mouse heart function in Figure 1 showed overall reduced systolic function in LmnaG609G/wt progeria mice compared to wt mice. However, the mouse size between different genotypes should be taken into consideration before drawing such conclusion as these progeria mice are significantly smaller at 38-week age than wt mice (A Zaghini, et al., 2020 Exp Gerontology) although Progerinin might help maintaining their body weight over 12-week treatment period. 

2. Previous studies have reported abnormal LV diastolic physiology with normal systolic function in LmnaG609G/G609G homozygous mice (SI Murtada, et al., 2020 J Royal Society Interface) and in HGPS patients (A Prakash, et al., 2018 JAMA Cardiology). In current study, progeria mice showed decreased diastolic function at late stage close to their endpoint (50-week) assessed by only measuring LV wall strain. To sufficient support this conclusion, a more comprehensive evaluation should be included, i.e., measuring mitral valve inflow velocity at PW doppler mode, and mitral annular velocity at PW tissue doppler mode under apical 4-chamber view. 

3. Was there any gender difference for cardiac dysfunction or improvement among 4 cohorts? Why did the authors only use male mice in this study?

4. In Figure 3A, it’s not convincing to claim the nuclear morphological improvement upon Progerinin treatment by a few blurry images without showing clear boundary of nucleus and without quantitative analysis. Immunofluorescence staining with anti-Lamin A/C antibody on frozen heart sections can better trace nuclear shape than conventional chromogenic staining on FFPE slides. 

5. Have the authors examined the heart fibrosis by histology study after Progerinin treatment?

6. Progerinin (or its previous version JH4) was designed as a Lamin A-progerin binding blocker. It also showed reduction of progerin expression in current study and the previous report by the same research group. What is the mechanism of progerin decrease through inhibition of Lamin A-progerin binding by Progerinin? 

7. Oral administration of Progerinin in diet is generally friendly to mice but it can’t tell how much dosage each mouse actually consumes each day.  Alternative approach, i.e., intraperitoneally injection with precise dosing, would provide valuable information in terms of future clinical trial on HGPS patients.

Other comments:

1) What is the unit for Y-axis in Figure 2S, gram or percent of body weight change? Please add body weight data over the 12-week treatment period that would be much clear for readers.

2) What is the protein in the middle panel of Figure 3B WB?  If that is the similar one to the top panel but with shorter exposure time, a side note can be added or indicated in figure legend.

3) The method of cardiac echo study should be described clearly including what parameters were acquired at which modalities and planes, i.e., B-mode or M-mode; PSLAX or PSSAX, etc. What are the settings that LV wall PW tissue doppler was conducted for diastolic physiology measurement in in Figure 1D. What is the software's name for echo data analysis (Vevo Lab)?

4) The supplemental figure legends are missing.

Author Response

Thank you so much for taking your time to review our paper.

  1. The echocardiographic evaluation of mouse heart function in Figure 1 showed overall reduced systolic function in LmnaG609G/wt progeria mice compared to wt mice. However, the mouse size between different genotypes should be taken into consideration before drawing such conclusion as these progeria mice are significantly smaller at 38-week age than wt mice (A Zaghini, et al., 2020 Exp Gerontology) although Progerinin might help maintaining their body weight over 12-week treatment period. 

→Thank you for your important point. Change of body weight can be related to the systolic function. Consideration of the relation of body size and systolic function between different genotypes is important. However, in this study, we mainly focus on the result that maintaining body weight by Progerinin improves overall systolic function in progeria mouse model. Reflecting your opinion, we updated our conclusion in the Discussion section.

  1. Previous studies have reported abnormal LV diastolic physiology with normal systolic function in LmnaG609G/G609Ghomozygous mice (SI Murtada, et al., 2020 J Royal Society Interface) and in HGPS patients (A Prakash, et al., 2018 JAMA Cardiology). In current study, progeria mice showed decreased diastolic function at late stage close to their endpoint (50-week) assessed by only measuring LV wall strain. To sufficient support this conclusion, a more comprehensive evaluation should be included, i.e., measuring mitral valve inflow velocity at PW doppler mode, and mitral annular velocity at PW tissue doppler mode under apical 4-chamber view. 

→ Thank you for your important suggestion. However, in this study, we could not obtain the data you mentioned. We tried to but hard to obtain because of our systemic problems.

→ Reflecting your opinion, we are considering to assess more comprehensive evaluations in our future study. We are planning to evaluate the cardiac function and physiology between homozygous LmnaG609G/G609G mice and heterozygous LmnaG609G/WT mice by treatment with Progerinin.

  1. Was there any gender difference for cardiac dysfunction or improvement among 4 cohorts? Why did the authors only use male mice in this study?

→In this study, we only use male mice. Because it is known that change in cardiac function and structure are more significant in males than females (Zhang, et al., 2021, Sci Rep; Koch, et al., 2013, Ultrasound Med Biol). In this study, we did not compare the gender difference.

→ However, we also plan to compare the gender difference in our future study.

  1. In Figure 3A, it’s not convincing to claim the nuclear morphological improvement upon Progerinin treatment by a few blurry images without showing clear boundary of nucleus and without quantitative analysis. Immunofluorescence staining with anti-Lamin A/C antibody on frozen heart sections can better trace nuclear shape than conventional chromogenic staining on FFPE slides.

→We changed the improved images and also added a graph of percentage of abnormal nuclei in supplementary data (Supple Figure 7B).

→ Thank you so much for your comments. Reflecting your suggestion, we will use frozen heart sections on the next study.

  1. Have the authors examined the heart fibrosis by histology study after Progerinin treatment?

→We have already examined the heart fibrosis by histology study after Progerinin treatment in our previous study (Kang, et al., 2021 Commun Biol). In this study, we only examined the aortic fibrosis.

  1. Progerinin (or its previous version JH4) was designed as a Lamin A-progerin binding blocker. It also showed reduction of progerin expression in current study and the previous report by the same research group. What is the mechanism of progerin decrease through inhibition of Lamin A-progerin binding by Progerinin? 

→In our previous study, we discovered that progerin alone did not induce nuclear deformation in lamin A/C-deficient neuron or embryonic neuron cells (Lee et al, J Clin Invest,2016). It seems that progerin can be stable under existence of lamin A by directly binding to lamin A. Up to now, we figured out that separating progerin from Lamin A induced the downregulation of progerin expression. In present, we are studying the detailed mechanism of progerin elimination after inhibition of Lamin A-progerin.

  1. Oral administration of Progerinin in diet is generally friendly to mice but it can’t tell how much dosage each mouse actually consumes each day.  Alternative approach, i.e., intraperitoneally injection with precise dosing, would provide valuable information in terms of future clinical trial on HGPS patients.

→We have already examined by IP injection, oral gavage, and also using diet pellets in our previous study. And we confirmed that oral administration of Progerinin by diet pellets was more effective than IP injection in previous study (Kang et al., Commun Biol, 2021).

→Although we know that Progerinin dosing by IP injection or using oral gavage is more precise than diet pellets, in this study, we tried to let the mice not to be stressed. Because frequent anesthesia can also be stressful to the mice. So, we chose the most stressless method for treatment with Progerinin.

Other comments:

1) What is the unit for Y-axis in Figure 2S, gram or percent of body weight change? Please add body weight data over the 12-week treatment period that would be much clear for readers.

→ We are sorry for the lack of information. Y-axis represents percent of body weight change. As you recommended, we changed body weight data over the 12-week treatment period.

2) What is the protein in the middle panel of Figure 3B WB?  If that is the similar one to the top panel but with shorter exposure time, a side note can be added or indicated in figure legend.

→We are very sorry for confusing you. We noticed that we made a mistake in Figure 3B. We have added two different cropped ‘PG’ bands. We have done WB repeatedly. Each of two bands is from separated experiment. We corrected Figure 3B.

3) The method of cardiac echo study should be described clearly including what parameters were acquired at which modalities and planes, i.e., B-mode or M-mode; PSLAX or PSSAX, etc. What are the settings that LV wall PW tissue doppler was conducted for diastolic physiology measurement in in Figure 1D. What is the software's name for echo data analysis (Vevo Lab)?

→ We updated the method of echocardiography and cardiac function analysis (part 2.6.).

→ The measurement of DWS was calculated by using the following formula: (LVPWs-LVPWd)/LVPWs*100%.

→We used ‘Vevo Strain Software’.

4) The supplemental figure legends are missing.

→The supplemental figure legends are bottom of the each slide.

We greatly appreciate the time you spent to us.

Reviewer 4 Report

In this paper the authors describe the improvement of cardiac abnormalities in a model mouse of the Hutchinson-Gilford progeria syndrome (HGPS) or progeria induced by the administration of progerinin, a progerin-binding compound.

Progeria is a rare disease fro which only a drug has been approved and has limited efficacy, so new therapeutic approaches are needed. Progeria is caused by a point mutation in the lamin A gen that causes the production of an abnormal form of the prelamin A that is called progerin. This protein accumulates in the nuclear membrane producing extension malfunctioning that affects to all peripheral organs and finally causes the death of the individual at an average age of 15 years. The main cause of death is related with cardiovascular alterations, so any potential treatment needs to improve cardiovascular parameters such as the ones described in this paper as well as to decrease the levels of progerin in aorta and heart tissue. Hence, the research topic is important and of current interest. The manuscript is clearly written, the methodology is technically sound and the conclusions are supported by the data. However, some aspects need to be addressed before the work can be published in Cells journal.

Major modifications required:

1) In vivo dose effectively administrated of progerinin should be indicated.

2) Progerinin source and standard quality control data of the compound should be provided. In addition, preparation of the solution in monoololein, source of monoolein, purity criteria for progerinin, number of batches used, reproducibility among them… should be provided.

3) In vivo levels of progerinin in tissues of interest and complete in vivo pharmacokinetics should be provided.

4) One limitation of the present study is the use of heterozygous LmnaG609G/WT mice instead of homozygous LmnaG609G/G609G mice, being this last one the real model of the disease. The use of heterozygous instead of homozygous should be justified and the inherent limitations discussed.

5) In Supplementary Figure 2, it is not clear the magnitude represented in the y axis. What does it mean change from baseline? Is it percentage? Or absolute grams? This should be explained.

6) Cardicac phenotype improvement should be discussed in comparison with the effects induced by lonafarnib (the only drug currently in the market approved for progeria treatment) as well as with other described therapeutic approaches (Nature 2021, 589, 608; Elife 2021, 10, e63284; ACS Cent. Sci. 2021, 7, 1300; Nat. Med. 2021, 27, 526; Nat. Med. 2021, 27, 536).

Minor points:

-In the abstract, line 16, “aberrant splicing LMNA gene” should be “aberrant splicing of the LMNA gene”

Author Response

Thank you so much for taking your time to review our paper.

Major modifications required:

1) In vivo dose effectively administrated of progerinin should be indicated.

→ We are sorry for the lack of information. We updated in the method section (part 2.5.).

2) Progerinin source and standard quality control data of the compound should be provided. In addition, preparation of the solution in monoololein, source of monoolein, purity criteria for progerinin, number of batches used, reproducibility among them… should be provided.

→ The information of Progerinin is well described in our previous study. In present study, we mentioned it in method section shortly.

→ We added the information of monoolein-based solution in the method section (part 2.4.)

3) In vivo levels of progerinin in tissues of interest and complete in vivo pharmacokinetics should be provided.

→ We have already provide the data in our previous study. We mentioned it shortly in material and method sections.

4) One limitation of the present study is the use of heterozygous LmnaG609G/WT mice instead of homozygous LmnaG609G/G609G mice, being this last one the real model of the disease. The use of heterozygous instead of homozygous should be justified and the inherent limitations discussed.
→ As you mentioned, homozygous LmnaG609G/G609G mouse is better model for HGPS study. However, there was difficulty in producing enough homozygous individuals at the same time. We conducted our experiments with 38-week-old heterozygous LmnaG609G/WT mice as these progeria mice are showing noticeable aging phenotypes at that time. Even more, frequent anesthesia can affect to the survival of homozygous LmnaG609G/G609G mice. We have sometimes observed that homozygous LmnaG609G/G609G mice did not wake up after anesthesia. That’s why we selected heterozygous LmnaG609G/WT mice for analyzing cardiac function in this study. We addressed this limitation in the discussion section.

→ Reflecting your opinion, we are considering to compare the cardiac function and physiology of homozygous LmnaG609G/G609G mice and heterozygous LmnaG609G/WT mice in our future study.

5) In Supplementary Figure 2, it is not clear the magnitude represented in the y axis. What does it mean change from baseline? Is it percentage? Or absolute grams? This should be explained.

→ I am sorry for confusing you. The y axis represents percentage change in body weight from baseline (from start-point to end-point). I have corrected the graph clearly. And we exchanged the previous graph into the body weight data over the 12-week treatment period to be clear.

6) Cardiac phenotype improvement should be discussed in comparison with the effects induced by lonafarnib (the only drug currently in the market approved for progeria treatment) as well as with other described therapeutic approaches (Nature 2021, 589, 608; Elife 2021, 10, e63284; ACS Cent. Sci. 2021, 7, 1300; Nat. Med. 2021, 27, 526; Nat. Med. 2021, 27, 536).

→ Thank you for your important point. Lonafarnib also improves arterial structure and left ventricular diastolic function (Elife 2023, 82728). Both of Progerinin and lonafarnib shows favorable effect in cardiac phenotypes in LmnaG609G mice. However, we showed that treatment with Progerinin is more non-toxic and effective than lonafarnib in our previous study. Progerinin maintained the body weight of LmnaG609G mice and extended life span of LmnaG609G mice longer than lonarfarnib. Based on previous and present results, we suggest that Progerinin can be a more promising treatment for HGPS. We addressed this point in the discussion section.

→ We know that other therapeutic approaches also improve cardiac phenotype in progeria model mice. However, we think that it is not proper to compare the effects between them in this study because of using different types of mouse model and different or insufficient data of their studies.

Minor points:

-In the abstract, line 16, “aberrant splicing LMNA gene” should be “aberrant splicing of the LMNA gene”

→ Thank you very much for your correction. We edited in the abstract.

We greatly appreciate the time you spent to us.

Round 2

Reviewer 1 Report

1.     In Suppl. Fig. 2, there are no statistical marks in the bar graph. Please identify which data points are significantly different when comparing the Vehicle and Treated groups (to support the P < 0.05 in the figure legend). Also, body weights at 38-weeks (the start of the treatment) should be reported.

2.     In Fig. 2C, the authors performed one-way ANOVA statistics, which is correct. However, they do not report whether the Vehicle and Treated groups are statistically different. Please report whether the two groups are significantly different in the figure.

3.     In Fig. 1, the authors compare Vehicle-treated wild type mice with Vehicle- and Treated-progeria mice by the t-test. The groups should be analyzed by ANOVA. Analyze the data by ANOVA statistics and report which groups are statistically different in the figure. Update the manuscript as needed.

Author Response

We really appreciate the time you spent reviewing our paper.

1. In Suppl. Fig. 2, there are no statistical marks in the bar graph. Please identify which data points are significantly different when comparing the Vehicle and Treated groups (to support the P < 0.05 in the figure legend). Also, body weights at 38-weeks (the start of the treatment) should be reported.

- As you mentioned, we added statistical marks in the bar graph. We also reported the average amounts of actual body weight at the start-point in the supplementary figure legend.

2. In Fig. 2C, the authors performed one-way ANOVA statistics, which is correct. However, they do not report whether the Vehicle and Treated groups are statistically different. Please report whether the two groups are significantly different in the figure.

- We added the result of statistical analysis between the vehicle and treated groups using one-way ANOVA statistics.

3. In Fig. 1, the authors compare Vehicle-treated wild type mice with Vehicle- and Treated-progeria mice by the t-test. The groups should be analyzed by ANOVA. Analyze the data by ANOVA statistics and report which groups are statistically different in the figure. Update the manuscript as needed.

- We re-analyzed the data using two-way ANOVA statistics (Figure 1 and Supple Figure 3) and also updated the manuscript.

Thank you so much.

Reviewer 3 Report

The authors made edits and changes to this manuscript that significantly improved its quality for publication. 

Please indicate the total nuclei number counted in each group in Figure 7S.

Please check typos and correct grammar errors, i.e., delete "to" after "affect" in line 271.

Progerinin blocks the binding between progerin and Lamin A/C that could potentially increase the release of this toxic protein from nuclear inner envelope thus might facilitate its degradation in cytosol. Looking forward to seeing the authors' new findings about its detailed mechanism. 

Author Response

We really appreciate the time you spent reviewing our paper.

Please indicate the total nuclei number counted in each group in Figure 7S.

- We indicated the total nuclei number (300 nuclei/sample) in Figure 7S.

Please check typos and correct grammar errors, i.e., delete "to" after "affect" in line 271.

- Thank you so much for your correction. We edited and checked other grammar errors.

Progerinin blocks the binding between progerin and Lamin A/C that could potentially increase the release of this toxic protein from nuclear inner envelope thus might facilitate its degradation in cytosol. Looking forward to seeing the authors' new findings about its detailed mechanism. 

Thank you very much for your comments.

Reviewer 4 Report

The authors have addressed some of my concerns but not of all them. Significant aspects that should be addressed are:

1) Progerinin source and quality control data of the compound should be provided. It is ok to cite the previous work where progerinin was originally described, but the authors should provide specific data of the source of progerinin used to carry out the reported work. Since performing a set of in vivo experiments as the one described by the authors involve the synthesis of a significant amount of compound, authors should indicate whether the compound was synthesized by them and in which scale or if it was subcontracted. They should indicate whether all the compound was synthesized in a batch or in several batches and they should provide HPLC/MS traces confirming at least a purity of 95%. This is important because ensuring the purity of compound is of utmost important for supporting the validity of the obtained data.

2) Compound concentration in heart after oral administration at relevant time points should be provided (or explicitly cited if already described in their previous paper where progerinin was originally described). It is not correct to mention a general sentence saying that PK studies where already carried out in the previous paper. It is important to confirm the levels of the compounds reached at the site of action (heart) since all the discussion assumes that it is the presence of the compound and its binding to progerin the responsible of the observed effects. Then, for the sake of the argumentation, levels of the compound in the tissue of interest should be indicated.  

3) Like mentioned in my previous revision, one limitation of the present study is the use of heterozygous LmnaG609G/WT mice instead of homozygous LmnaG609G/G609G mice, being this last one the real model of the disease. The use of heterozygous instead of homozygous should be justified and the inherent limitations discussed. The authors say that they have addressed this limitation in the discussion section, but I have not found any paragraph that explicitly indicates this limitation nor any reference of the homozygous model vs the heterozygous model. The authors should explicitly address this limitation.

Minor points:

1) Numbers of mice used in the different experiments should be explicitly indicated in the figure legends.

Author Response

We really appreciate the time you spent reviewing our paper.

1) Progerinin source and quality control data of the compound should be provided. It is ok to cite the previous work where progerinin was originally described, but the authors should provide specific data of the source of progerinin used to carry out the reported work. Since performing a set of in vivo experiments as the one described by the authors involve the synthesis of a significant amount of compound, authors should indicate whether the compound was synthesized by them and in which scale or if it was subcontracted. They should indicate whether all the compound was synthesized in a batch or in several batches and they should provide HPLC/MS traces confirming at least a purity of 95%. This is important because ensuring the purity of compound is of utmost important for supporting the validity of the obtained data.

- We updated the section of the 'Progerinin synthesis and characterization data' in Methods. The source of Progerinin is decursinol, which is isolated from the roots of Angelica gigas Nakai. A one-step synthesis reaction of coupling the decursinol and side chains was carried out to produce Progerinin. And we used the same batch of Progerinin which had been already used in our previous study. We updated what you mentioned.

2) Compound concentration in heart after oral administration at relevant time points should be provided (or explicitly cited if already described in their previous paper where progerinin was originally described). It is not correct to mention a general sentence saying that PK studies where already carried out in the previous paper. It is important to confirm the levels of the compounds reached at the site of action (heart) since all the discussion assumes that it is the presence of the compound and its binding to progerin the responsible of the observed effects. Then, for the sake of the argumentation, levels of the compound in the tissue of interest should be indicated.  

- As you mentioned, we think it is important to measure the level of the compound in the site of action (heart). But we only measured the compound level in the blood plasma. We think the compound might work effectively on the heart since the compound moves through the blood to the tissues. Unfortunately, there are no confirmed results in this study. However, since the issue you mentioned is a very important point, we are planning to analyze the actual level of the compound in each organ in our future study.

3) Like mentioned in my previous revision, one limitation of the present study is the use of heterozygous LmnaG609G/WT mice instead of homozygous LmnaG609G/G609G mice, being this last one the real model of the disease. The use of heterozygous instead of homozygous should be justified and the inherent limitations discussed. The authors say that they have addressed this limitation in the discussion section, but I have not found any paragraph that explicitly indicates this limitation nor any reference of the homozygous model vs the heterozygous model. The authors should explicitly address this limitation.

- We are sorry about this. We missed the statement in the revised paper by mistake. We added in the Discussion section (line 277) and checked again precisely this time.

Minor points:

1) Numbers of mice used in the different experiments should be explicitly indicated in the figure legends.

- We indicated the numbers of mice in each figure legend.

Thank you very much again.